# Genomic Characterization of Rare Primary Cardiac Sarcoma Entities

**DOI:** 10.3390/diagnostics13020214

**Published:** 2023-01-06

**Authors:** Livia Gozzellino, Margherita Nannini, Carmine Pizzi, Ornella Leone, Barbara Corti, Valentina Indio, Chiara Baldovini, Pasquale Paolisso, Alberto Foà, Davide Pacini, Gianluca Folesani, Angela Schipani, Alice Costa, Gianandrea Pasquinelli, Maria Abbondanza Pantaleo, Annalisa Astolfi

**Affiliations:** 1Department of Experimental, Diagnostic and Specialty Medicine (DIMES), Alma Mater Studiorum, University of Bologna, 40138 Bologna, Italy; 2Division of Oncology, IRCCS Azienda Ospedaliero, Universitaria di Bologna, 40138 Bologna, Italy; 3Department of Experimental Diagnostic and Specialty Medicine, IRCCS Azienda Ospedaliero-Universitaria, 40138 Bologna, Italy; 4Division of Pathology, Cardiovascular and Cardiac Transplant Pathology Unit, IRCCS Azienda Ospedaliero-Universitaria, 40138 Bologna, Italy; 5Department of Veterinary Medical Sciences, Alma Mater Studiorum, University of Bologna, 40138 Bologna, Italy; 6Cardiac Surgery Unit, IRCCS Azienda Ospedaliero-Universitaria, 40138 Bologna, Italy; 7Department of Pharmaceutics, Utrecht Institute for Pharmaceutical Sciences, Utrecht University, 3584 CS Utrecht, The Netherlands; 8Division of Pathology, IRCCS Azienda Ospedaliero, Universitaria di Bologna, 40138 Bologna, Italy

**Keywords:** cardiac sarcomas, leiomyosarcoma, osteosarcoma, whole-transcriptome sequencing, bioinformatics

## Abstract

Primary cardiac sarcomas are considered rare malignant entities associated with poor prognosis. In fact, knowledge regarding their gene signature and possible treatments is still limited. In our study, whole-transcriptome sequencing on formalin-fixed paraffin-embedded (FFPE) samples from one cardiac osteosarcoma and one cardiac leiomyosarcoma was performed, to investigate their mutational profiles and to highlight differences and/or similarities to other cardiac histotypes. Both cases have been deeply detailed from a pathological point of view. The osteosarcoma sample presented mutations involving *ATRX*, *ERCC5*, and *COL1A1*, while the leiomyosarcoma case showed *EXT2*, *DNM2*, and *PSIP1* alterations. Altered genes, along with the most differentially expressed genes in the leiomyosarcoma or osteosarcoma sample versus the cardiac angiosarcomas and intimal sarcomas (e.g., *YAF2*, *PAK5*, and *CRABP1*), appeared to be associated with cell growth, proliferation, apoptosis, and the repair of DNA damage, which are key mechanisms involved in tumorigenesis. Moreover, a distinct gene expression profile was detected in the osteosarcoma sample when compared to other cardiac sarcomas. For instance, *WIF1,* a marker of osteoblastic differentiation, was upregulated in our bone tumor. These findings pave the way for further studies on these entities, in order to identify targeted therapies and, therefore, improve patients’ prognoses.

## 1. Introduction

Primary cardiac sarcomas are extremely rare malignant tumors of the heart, representing ~1% of all soft-tissue sarcomas and comprising a heterogeneous spectrum of histotypes [1,2]. They are classified according to their cardiac site of origin (cardiac chambers, pericardium, vena cava, and aorta) and intimal or mural localization. Angiosarcoma is the most common histotype, with an incidence of 7.3–8.5% among all variants, followed by undifferentiated high-grade pleomorphic cardiac sarcomas, with an incidence of 4.1–8.5%. Leiomyosarcomas, rhabdomyosarcomas, synovial sarcomas, liposarcomas, fibrosarcomas, myxoid fibrosarcomas, osteosarcomas, and chondrosarcomas represent rarer histological entities. Angiosarcomas are predominantly right-sided and are usually located in the right atrial cavity, while poorly differentiated cardiac sarcomas as well as other rarer histological subtypes are mostly located in the left atrial cavity.

The overall prognosis of primary cardiac sarcomas is generally poor and mainly correlates with tumor localization and with the involvement of the surrounding vital structures. These factors negatively affect surgical radicality and enhance the biological aggressiveness of most histotypes [3,4,5].

To date, clinical management of primary cardiac sarcomas has been complex and challenging, from diagnostics, often requiring a multimodality approach, to medical therapies, which still play a modest role [5]. This complexity is also due to the molecular background, which is extremely heterogeneous, though related knowledge is still very limited [6]. The molecular alterations detected in cardiac angiosarcomas often involve *KDR*, *KIT*, and the homozygous deletion of *CDKN2A*, while *KRAS* p.G13S and p.Q61K mutations have been reported in a few cases. As expected, cardiac undifferentiated pleomorphic sarcomas also have a complex genomic profile, including *MDM2* and *PDGFRA* amplification, *EGFR* aneuploidy or amplification, and *PDGFRB* mutations. Conversely, limited molecular data on rarer histotypes are available in the literature. In the past, a primary cardiac leiomyosarcoma harboring a *HRAS* mutation was described [7]. Furthermore, in a single case of primary cardiac osteosarcoma, allelic losses or imbalances involving 1p36, 9p22, 10q23, 17q22, and 22q13 have been reported [8]. More recently, a whole-transcriptome sequencing analysis of a cardiac myxofibrosarcoma revealed a gene expression profile similar to that of angiosarcomas [9].

From our clinical series of adult primary cardiac sarcomas, herein we report and focus on two rare entities with transcriptome profiles that have been investigated with the aim of better describing these histotypes and, consequently, of identifying potential targets for more personalized therapies.

## 2. Materials and Methods

### 2.1. Patients

From 2005 to 2017, 11/358 patients with primary cardiac sarcomas were identified in our sarcoma population. Mean age was 51.8 years (range = 35–73). Main patients’ features are listed in Table 1. Histologically, the case series was extremely heterogenous, with an expected prevalence of pulmonary artery intimal sarcomas (5/11—45%) and angiosarcomas (3/11—27%). Other rare subtypes including an osteosarcoma, a myxofibrosarcoma, and a leiomyosarcoma were also identified. Almost all patients were metastatic at the time of diagnosis, presenting an aggressive clinical course.

### 2.2. Coding Transcriptome Sequencing

FFPE slides were reviewed by an expert pathologist and manually macrodissected to obtain an enrichment of tumor tissue of at least 70%. Total RNA was extracted using Recover. All Total Nucleic Acid Isolation Kit (Thermo Fisher Scientific, Waltham, MA, United States), and cDNA libraries were synthesized from 100 ng total RNA using TruSeq RNA Exome kit (Illumina, San Diego, CA, United States), in accordance with the instructions of the manufacturers. Briefly, cDNA libraries were synthesized from fragmented RNA, ligated to sequencing adapters, and amplified; then coding exon sequences were captured by hybridization to a pool of exonic probes. Libraries were sized with Agilent DNA 7500 chips on the Bioanalyzer 2100 (Agilent Technologies, Taiwan) and quantified with a fluorometric assay (Quant-IT Picogreen assay; Life Technologies, Carlsbad, CA, United States). Paired-end libraries (SRA Accession Number: PRJNA896891) were sequenced at 80 bp on a NextSeq500 instrument (Illumina, San Diego, CA, United States).

### 2.3. Bioinformatic Analysis

Our bioinformatic workflow was based on a primary quality check with FastQC and MultiQC to ensure read quality was acceptable and transcriptomes were comparable among samples [10,11]. Secondly, reads were mapped on the human genome to identify expressed genes. To determine transcript abundance, read quantification was performed obtaining raw counts. Subsequently, to ensure comparability, raw counts were normalized as counts per million (CPM) and transcripts per million (TPM), which were required for differential gene expression (DGE) and principal component analysis (PCA), respectively.

In detail, STAR was adopted to map paired reads on the reference human genome hg38 [12]; while duplicates’ removal, sorting, and indexing were performed with Samtools [13]. Gene expression was quantified and normalized as CPM and TPM, after obtaining raw gene counts with the Python package HTseq-count [14]. Starting from CPM, differentially expressed genes (*q*-value < 0.05, FDR-corrected) between either the leiomyosarcoma or osteosarcoma case and the group including angiosarcomas (CS6 and CS7) and intimal sarcomas (CS1, CS2, CS3, CS4 and CS5) were identified with the R-bioconductor package edgeR [15]. Using the R package prcomp [16] instead, TPM were employed to perform PCA, which also included intimal sarcomas (n = 5) and angiosarcomas (n = 2).

Additionally, gene fusion detection and small variant calling combined with variant annotation via Nirvana were performed on BaseSpace, an Illumina webtool (http://euc1.sh.basespace.illumina.com/ accessed on 4 June 2022). For gene-fusion detection, protein-coding and lncRNA genes were included to reduce false positives.

The vcf output files were filtered to include only stop gain/splice donor/splice acceptor/splice region/frameshift indels/inframe deletion/inframe insertion/initiator codon/ATG loss/missense mutations predicted as deleterious or probably damaging by SIFT predictor, with a GnomAD Frequency < 0.01, a QC metric quality ≥ 30, a total read depth > 14, an alt allele depth > 3, and a variant read frequency > 0.15. Finally, only genes belonging to Tier 1 of the Cancer Gene Census list (https://cancer.sanger.ac.uk/census accessed on 4 June 2022) were considered for further analysis. Sequencing data of both samples were analyzed using the DRAGEN RNA app (version 3.10.4).

### 2.4. Sanger Sequencing

To confirm the presence of alterations detected by bioinformatic analyses on genomic DNA and to ensure they were not due to RNA editing, DNA was extracted from manually macrodissected FFPE slides using the QIAmp DNA Micro KIT (Qiagen, Hilden, Germany), while cDNA was synthesized from 1 μg of total RNA using the High-Capacity cDNA Reverse Transcription Kit (Thermo Fisher Scientific, Waltham, MA, USA), in accordance with the instructions of the manufacturers.

Primers flanking the sequences of interest were designed using Primer3 Plus (https://www.bioinformatics.nl/cgi-bin/primer3plus/primer3plus.cgi accessed on 30 June 2022), either on genomic DNA or cDNA, and quality was checked by Blast alignment. Primer sequences are available upon request.

Lastly, to confirm point mutations and fusion transcripts, PCR amplification was performed on 40 ng of genomic DNA or 200 ng of cDNA using FastStart Taq Polymerase (Roche Diagnostics), with an annealing of 58 °C. Amplified fragments were sequenced on both strands using the Big Dye Terminator v1.1 Cycle Sequencing kit (Applied Biosystems) on ABI 3730 Genetic Analyzer (Applied Biosystems).

## 3. Results

Due to the rarity of cardiac sarcomas, we present the detailed results from the transcriptome analysis of two unusual histotypes: an osteosarcoma (CS9) and a leiomyosarcoma (CS11).

### 3.1. Patients

Case 1: In March 2009, a 39-year-old male presented dyspnea and tachycardia, associated with recurrent lipotimia episodes. Thoracic angio-contrast tomography (CT), cardiac catheterization, angiography, and echocardiography revealed a vascularized left atrial mass. Thus, in April 2009, the patient underwent surgical resection. Macroscopically, the mass appeared partially encapsulated with a diameter of 9 cm and was formed by solid whitish-gray tissue (Figure 1A). In multifocal areas of the chondroid and osteoid matrices, a histological examination showed mixed spindle and round cells with solid growth pattern, prominent nuclear atypia, and high mitotic activity (21/10 high-power field—HPF). Some giant osteoclastic cells were also present. According to immunohistochemical analysis (IHC), the tumor cells were diffusely positive for vimentin and had a high multifocal expression of wide-spectrum cytokeratins. Morphological aspects, along with the IHC findings, were coherent with the diagnosis of osteosarcoma (Figure 1). Subsequently, the patient underwent adjuvant chemotherapy with methotrexate, cisplatin, adriamycin, and ifosfamide until February 2010. In March 2011, due to an extended and symptomatic local relapse, the patient underwent a left pneumonectomy, dying of the disease in June 2011.

Case 2: In December 2005, a 59-year-old female presented dyspnea and pleural effusion. Trans-esophageal echocardiography detected a pedunculated left atrial mass of a gelatinous consistency. Following surgical resection, a histological examination revealed spindle cells with thickened chromatin, prominent nuclear atypia, and high mitotic activity (10/10 HPF). Rounded cells with clear cytoplasm and multinuclear giant cells were present in delimited areas (Figure 2). Additionally, immunohistochemical analysis revealed that the tumor cells were diffusely positive for smooth muscle actin (SMA) and vimentin. The morphological aspects, along with the IHC findings, were coherent with the diagnosis of pleomorphic leiomyosarcoma. The patient died of the disease in August 2006.

### 3.2. Mutational Profile of Cardiac Osteosarcoma and Leiomyosarcoma

An unsupervised PCA showed that the osteosarcoma sample presents a distinct gene expression profile when compared to the other histotypes included in the analysis (leiomyosarcoma, cardiac angiosarcoma, and intimal sarcoma). Indeed, it is distinctly separated from them along the first principal component (Figure 3).

Supervised gene expression analysis of the leiomyosarcoma sample and the group comprising angiosarcomas and intimal sarcomas was characterized by 55 differentially expressed genes (*q*-value < 0.05) (Appendix A). The same analysis applied to the osteosarcoma case versus angiosarcomas and intimal sarcomas presented 482 differentially expressed genes (*q*-value < 0.05), confirming the uniqueness of the osteosarcoma expression profile (Appendix A). In both analyses, among the most differentially expressed genes, the biomarkers involved in major cell processes, such as cell division, histotype-specific differentiation, and DNA repair, were upregulated in both leiomyosarcoma and osteosarcoma (Figure 4). For instance, PAK5 and YAF2, which promote cell migration and invasion, were overexpressed in the leiomyosarcoma sample; while WIF1, a marker of osteoblastic differentiation, and DLK1, which is involved in cancer stem-like cell maintenance and cancer differentiation, were upregulated in the osteosarcoma sample.

Regarding gene mutations, we identified a total of 117 and 94 mutations in the osteosarcoma and leiomyosarcoma cases, respectively. Variants were classified according to their effect on the protein (Figure 5), resulting in a prevalence of missense and splice-site mutations. However, since our focus was on cancer genes, we restricted our analysis to mutations affecting the genes belonging to Tier 1 of the Cancer Gene Census list. This led to the identification of six alterations in the osteosarcoma sample and two in the leiomyosarcoma one.

In the osteosarcoma case, the ATRX gene was identified as carrying a frameshift mutation (c.1283_1284insT) affecting exon 9 on chr23:77683973 (Table 2). This ATRX variant was also validated by Sanger sequencing and determined the loss of the reading frame, introducing a premature stop codon at position 428 (Figure 6).

The ERCC5 gene presented a missense mutation (c.2636A>G) on chr13:102868215 (exon 12), where a Serine replaced an Asparagine at position 879 (Table 2). According to SIFT, this ERCC5 variant is deleterious and probably damaging according to PolyPhen. Moreover, on COSMIC, the alteration is classified as somatic, and its FATHMM prediction is pathogenic with a score of 0.99.

Lastly, the c.1461+2T>G variant affecting the COL1A1 gene seems to cause loss of the donor splice site at intron 21-22 on chr17:50194719. According to ClinVar, this alteration is likely pathogenic (Table 2).

In the leiomyosarcoma sample, a bioinformatic analysis of the coding transcriptome data detected an EXT2 gene missense mutation (c.1186G>A) located on exon 7 (Table 2), which was validated by Sanger sequencing (Figure 6). This variant led to the substitution of a Valine at position 396 by a Methionine. Additionally, this EXT2 alteration is classified as deleterious by SIFT, as it is probably damaging with a score of 0.99 by PolyPhen-2, and reported as likely benign for its association with osteochondroma in ClinVar.

The second alteration identified in the leiomyosarcoma sample was the c.688+2T>C variant affecting the DNM2 gene, which appears to lead to the loss of the donor splice site at exon 5 on chr19:10777218 (Table 2). DNM2 is linked to GTP hydrolyzation, endocytosis, and apoptotic cell maturation. The quality score of this alteration is 33, and the ACMG classification on Varsome is pathogenic.

Another noteworthy event in the leiomyosarcoma case was the fusion between the PSIP1 gene, involved in the DNA-binding transcription factor and chromatin-binding activity, and the STPG2 gene, which was validated by Sanger sequencing (Figure 6). This fusion transcript retained the reading frame and occurred between two intact exons: exon 2 of the PSIP1 gene located on chr9:15510117 and exon 3 of the STPG2 gene located on chr4:98128592 (Figure 7).

## 4. Discussion

Primary cardiac sarcomas comprise a rare and heterogeneous family of several histologies, characterized by varied clinical presentations and biological behaviors. Knowledge of the molecular background of these sarcomas, especially of the rarest histotypes, is still limited. Thus, we have studied the osteosarcoma and leiomyosarcoma cases by applying transcriptome sequencing analyses to enhance the understanding of these cardiac entities and to discover useful information for their clinical management and treatment.

Among the cardiac osteosarcoma alterations, the *ATRX* gene is noteworthy as it encodes a protein involved in transcription and chromatin remodeling [17,18]. It also plays a role in the ALT pathway, which administers telomere maintenance [18,19]. Furthermore, *ATRX* is usually mutated in 30% of osteosarcomas and is in fact considered to be one of the main drivers in osteosarcoma development, along with the following genes: *TP53*, *RB1*, *BRCA2*, *BAP1*, *RET*, *MUTYH*, *ATM*, *PTEN*, *WRN*, *RECQL4*, *FANCA*, *NUMA1*, and *MDC1* [20,21]. More specifically, point mutations in the *ATRX* gene seem to be associated with longer telomeres [21,22]. In some cases, *ATRX* depletion can lead to increased activity by the ALT pathway, which drives cancer cell immortalization [22].

The *ERCC5* gene encodes a single-stranded structure-specific DNA endonuclease involved in the nucleotide excision repair (NER) pathway [23,24]. Thus, since both *ATRX* and *ERCC5* are responsible for chromatin remodeling and DNA repair, mutations affecting these genes might alter DNA replication, leading to uncontrolled cell growth and proliferation.

Whilst it is established that cardiac leiomyosarcomas are rare and, therefore, lack a well-defined mutational profile, extra-cardiac leiomyosarcomas are better described in the literature. As proof, their most common signatures comprise the following genes: *TP53*, *RB1*, *PTEN*, *BRCA1/2*, *ATRX*, and *ATM* [25,26]. By contrast, germline mutations concerning the *EXT2* gene, which is involved in heparan sulphate biosynthesis and in tumor suppression, appear to be associated with hereditary multiple exostosis (HME), an autosomal dominant disease [27,28,29]. HME might lead to osteochondroma development, a benign bone tumor [28,30]. Additionally, these patients have an increased risk of developing chondrosarcomas and osteosarcomas [6,28].

Remarkably, the leiomyosarcoma sample we have analyzed did not present any *TP53*, *RB1*, and *ATRX* mutations, which instead characterize its histological class [31]. Homozygous deletions regarding *RB1*, *PTEN*, and *TP53*, which are common among leiomyosarcomas [25,32], could not be detected by this sequencing approach; nonetheless, we did not observe any strong evidence of gene expression downregulation.

Regarding the *PSIP1*–*STPG2* gene fusion, the former gene appears to be involved in cell growth, proliferation, and survival [33,34,35]. When upregulated, it enhances angiogenesis and prevents apoptosis in malignant cells, leading to aggressive cancer phenotypes [33]. Moreover, since *PSIP1* also plays a role in the recruitment of proteins involved in homology directed repair (HDR), when it is depleted, the number of DNA double-strand breaks (DSB) might increase [34,36]. In our case, only the first two exons of *PSIP1*, which consist mainly of untranslated regions, have been retained following gene fusion. Most likely, this leads to a loss of function by the PSIP1 protein and, subsequently, to genomic instability, which promotes cancer development. On the other hand, the *STPG2* breakpoint was at the beginning of exon 3; thus, only the first two exons, of which the first one is mainly composed of untranslated regions, were lost. This event might explain why *STPG2* was found to be upregulated among the most differentially expressed genes in the leiomyosarcoma case versus the angiosarcoma and intimal sarcoma cases.

In the leiomyosarcoma sample, the genes involved in cell growth, survival, and proliferation (*MFHAS1*, *YAF2*, *SOX5*, and *PAK5*) were also overexpressed. *MFHAS1* is an oncogene with expression in tumor-associated macrophages that has been associated with colorectal cancer (CRC) progression [37]. High expression of *YAF2* has been found in non-small cell lung cancer cells (NSCLCs), where it allows cell invasion and migration, preventing apoptosis [38]. *SOX5* seems to enhance epithelial mesenchymal transition (EMT), leading to metastasis in breast and prostate cancers [39]. *PAK5,* a binding protein of GTPases, has appeared to prevent drug-induced apoptosis in CRC cells, promoting cell proliferation. Thus, it has been selected as a potential therapeutic target [40].

Interestingly, in both leiomyosarcoma and osteosarcoma, an upregulated gene was *BMP5*, which encodes a secret ligand of the TFG-beta (transforming growth factor-beta) protein family and is associated with bone and cartilage development [41]. Additionally, it is usually overexpressed in synovial sarcomas. This event is remarkable mainly in the leiomyosarcoma sample, since, as previously mentioned, it presented *EXT2* alterations, which might contribute to osteochondroma onset. Therefore, this tumor sample has some features related to musculoskeletal malignancies.

In the osteosarcoma case versus the angiosarcomas and intimal sarcomas, the Wnt inhibitory factor 1 (*WIF1*) was also upregulated. *WIF1* is a marker of osteoblastic differentiation that highlights the specific differentiation lineage of this rare cardiac sarcoma [42]. It is usually overexpressed in the late stages of normal osteoblast maturation and in stratified osteosarcoma tumors. Moreover, upregulation concerned genes involved in cell proliferation, survival, differentiation, and migration (*DLK1* and *CRABP1*) [43,44]. *DLK1* expression characterizes drug-resistant tumors, since this gene allows cancer cells to maintain a stem-like phenotype [43]. High levels of *CRABP1* have been detected in synovial sarcomas, lymph node metastasis, and pancreatic neuroendocrine tumors (pNETs) [44]. Lastly, *COL11A2* overexpression is frequently associated with poor overall and event-free survival in patients diagnosed with osteosarcoma [45]. Consequently, these genes could be considered as markers or candidates for targeted therapies for this type of sarcoma.

The alterations identified in our analysis indicate an aggressive phenotype in both samples and suggest that these cases of cardiac osteosarcoma and leiomyosarcoma do not share common features with other types of cardiac sarcomas, including angiosarcomas, undifferentiated pleomorphic sarcomas, rhabdomyosarcomas, and myxofibrosarcomas [6]. For instance, our samples do not present any *POT1* and *TP53* mutations, unlike most cardiac angiosarcomas, any *PDGFRB* variants that characterize cardiac undifferentiated pleomorphic sarcomas, and any *KRAS* alterations commonly detected in cardiac rhabdomyosarcomas. However, according to the PCA results, angiosarcomas, intimal sarcomas, and leiomyosarcoma seem to share at least some features regarding their gene signature, unlike the osteosarcoma case, which instead appears to have a distinct pattern.

To date, the therapeutic landscape of both cardiac osteosarcomas and leiomyosarcomas has not differed from the one applied to sarcomas arising in more common primary sites, as previously stated by consensus conference guidelines [46,47]. However, the fact that the osteosarcoma greatly differs from the other histotypes, along with the presence of novel alterations in both cases (e.g., *ERCC5* and *PSIP1*), highlights the need to further explore these rare entities.

Since our sample size was limited, additional studies including more patients might help validating the events identified in our study, leading to multidisciplinary approaches and, perhaps, to individualized and molecular-based treatments. Another downside of our research is the lack of whole-exome sequencing (WES) analyses, where tumor samples are compared with their normal counterparts. Thus, future studies should include WES to better understand how gene amplifications and/or deletions can contribute to tumorigenesis, when combined with other mutational events. This might pave the way for more advanced combination therapies.

## 5. Conclusions

Ultimately, even if primary cardiac sarcomas still represent an unmet clinical need from several points of view, as we have learned from other rare diseases, sharing experiences remains the main option to improve our knowledge and patients’ treatment opportunities.

## Figures and Tables

**Figure 1 diagnostics-13-00214-f001:**
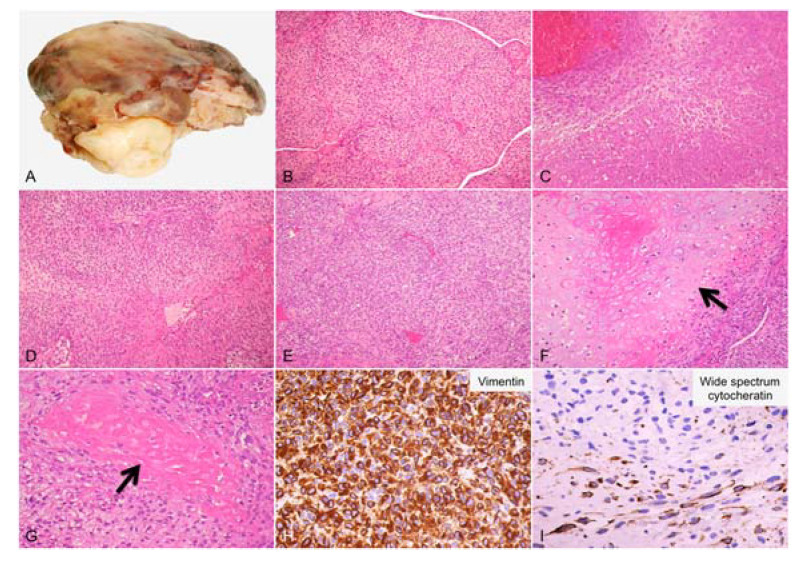
Histological findings representing the osteosarcoma sample. (**A**) Macroscopic aspect of the neoplastic mass formed by solid whitish-gray tissue; (**B**,**C**) malignant neoplasia forming solid nests or intersecting bundles (100×) with areas of necrosis (200×), respectively; (**D**–**G**) proliferation consisting of pleomorphic spindle (200×) or rounded cells (200×) with areas of chondroid (200×) and osteoid matrix (200×), respectively; (**H**,**I**) immunohistochemistry of neoplastic cells diffusely positive for vimentin (400×) and focally positive for wide-spectrum cytokeratin (400×), respectively. (**B**–**G**): Hematoxylin–eosin staining.

**Figure 2 diagnostics-13-00214-f002:**
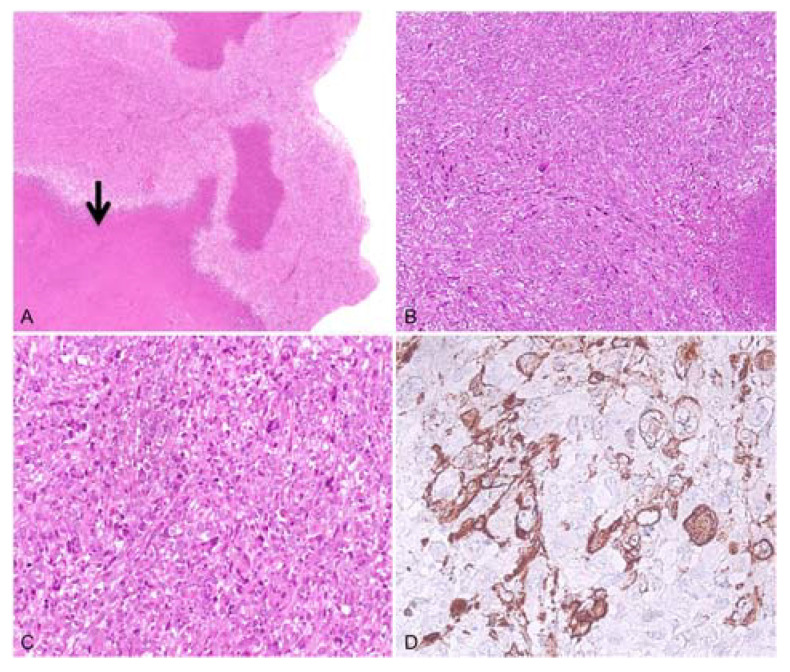
Histological findings representing the leiomyosarcoma case. (**A**,**B**) Malignant proliferation with extensive areas of necrosis (hematoxylin–eosin, 25×) prevalently made up of spindle cells with thickened chromatin (hematoxylin–eosin, 100×), respectively; (**C**) in delimited areas presence of some multinuclear giant cells and rounded cells with clear cytoplasm (hematoxylin–eosin, 200×); (**D**) immunohistochemistry of neoplastic cells diffusely positive for smooth muscle actin (400×).

**Figure 3 diagnostics-13-00214-f003:**
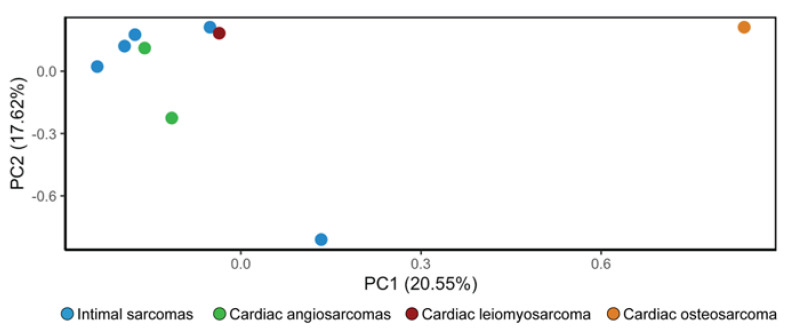
Unsupervised principal component analysis (PCA) performed on intimal sarcomas (n = 5), cardiac angiosarcomas (n = 2), leiomyosarcoma (n = 1), and osteosarcoma (n = 1). The osteosarcoma case (orange dot) is separated from the other samples along the first principal component, proving to have a distinct gene signature.

**Figure 4 diagnostics-13-00214-f004:**
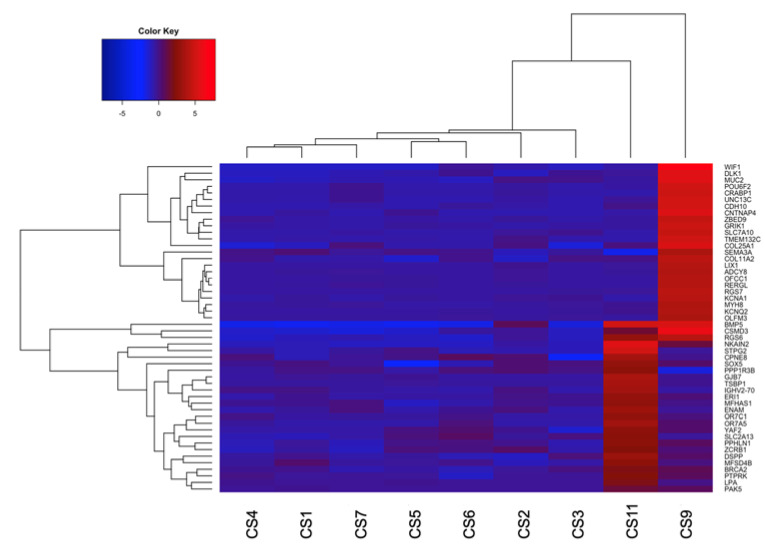
Hierarchical clustering representing the 25 most differentially expressed genes in the leiomyosarcoma sample (CS11) versus the group comprising angiosarcomas (CS6 and CS7) and intimal sarcomas (CS1, CS2, CS3, CS4 and CS5) and the 25 most differentially expressed genes in the osteosarcoma case (CS9) versus the angiosarcoma and the intimal sarcoma samples (*q*–value < 0.05).

**Figure 5 diagnostics-13-00214-f005:**
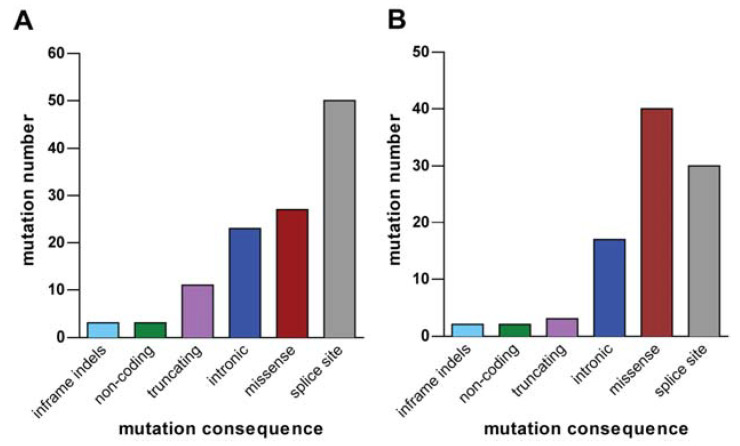
Histograms showing mutations present in the osteosarcoma (**A**) and in the leiomyosarcoma (**B**) cases, classified according to their effect on the protein.

**Figure 6 diagnostics-13-00214-f006:**
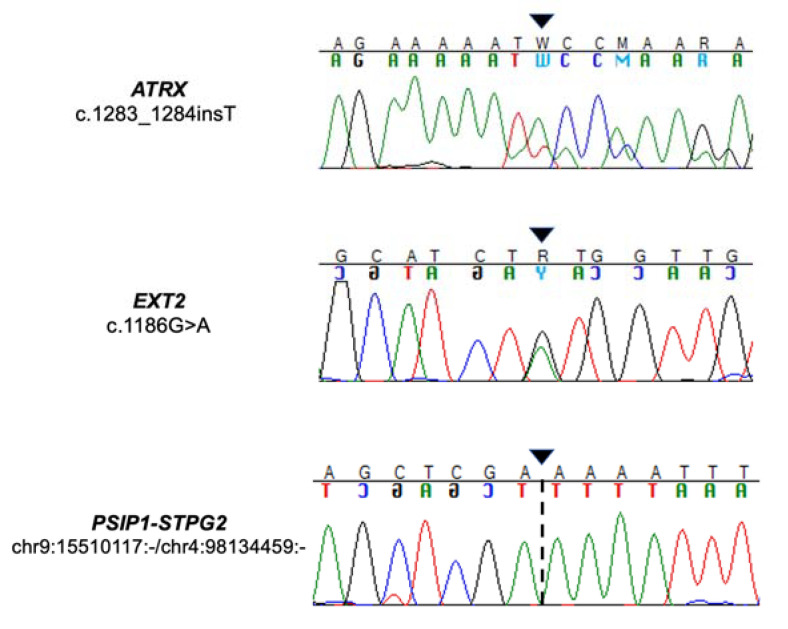
Chromatograms from Sanger sequencing analysis showing the following events (black arrows): in the osteosarcoma case, frameshift alteration in the ATRX gene due to the insertion of a thymine, and in the leiomyosarcoma case, missense mutation caused by the substitution of an adenine with a guanine in the EXT2 gene and PSIP1-STPG2 gene fusion, respectively.

**Figure 7 diagnostics-13-00214-f007:**
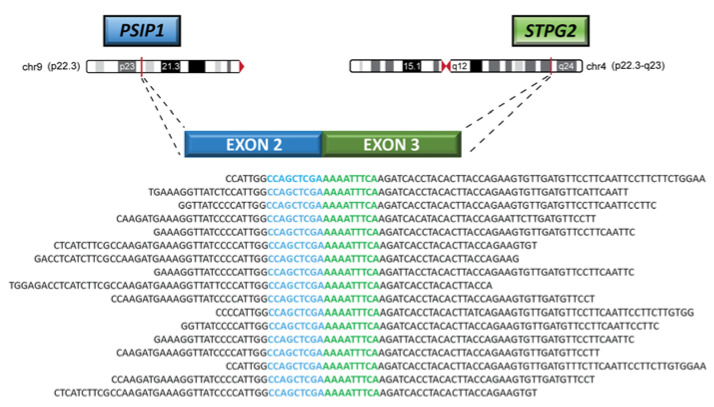
Representation of the PSIP1-STPG2 fusion transcript supported by reads mapped on the human reference genome hg38 and overlapping the fusion breakpoint. This event was derived from a rearrangement between chromosome 9 and chromosome 4 in the leiomyosarcoma case.

**Table 1 diagnostics-13-00214-t001:** Classification of our subset of patients (n = 11) diagnosed with primary cardiac sarcomas.

ID	Gender	Age	Histotype	Disease Status Dia	Last FU ^1^
CS1	M	50	pulmonary artery intimal sarcoma	Advanced	DOD ^2^
CS2	M	50	pulmonary artery intimal sarcoma	Advanced	DOD ^2^
CS3	F	45	pulmonary artery intimal sarcoma	Advanced	DOD ^2^
CS4	M	37	pulmonary artery intimal sarcoma	Advanced	DOD ^2^
CS5	F	69	pulmonary artery intimal sarcoma	Advanced	DOD ^2^
CS6	M	74	angiosarcoma	Advanced	DOD ^2^
CS7	M	39	angiosarcoma	Advanced	DOD ^2^
CS8	M	35	angiosarcoma	Localized	DOD ^2^
CS9	M	39	osteosarcoma	Advanced	DOD ^2^
CS10	F	73	myxofibrosarcoma	Localized	DOD ^2^
CS11	F	59	leiomyosarcoma	NA ^3^	NA ^3^

^1^ FU = follow-up; ^2^ DOD = died of disease; ^3^ NA = not available.

**Table 2 diagnostics-13-00214-t002:** Mutational analysis of gene variants in cardiac osteosarcoma (CS9) and leiomyosarcoma (CS11) cases.

ID	Gene	Chr	cDNA	Protein	Type	COSMIC	ClinVar	Varsome	gnomAD Frequency	Alt Allele Depth	Total Read Depth
CS9	*ATRX*	23	c.1283_1284ins(T)	p.(Asn428LysfsTer6)	frameshift	NA ^1^	NA ^1^	NA ^1^	NA ^1^	32	39
CS9	*ERCC5*	13	c.2636A>G	p.(Asn879Ser)	missense	pathogenic	benign	benign	0.00933	19	38
CS9	*COL1A1*	17	c.1461+2T>G	NA ^1^	spl. don. ^2^	NA ^1^	lik. path. ^3^	NA ^1^	NA ^1^	14	17
CS9	*USP6*	17	c.319G>T	p.(Gly107Cys)	missense	NA ^1^	NA ^1^	uncertain	0.0011	29	62
CS9	*FH*	1	c.172G>A	p.(Gly58Ser)	missense	NA ^1^	uncertain	NA ^1^	NA ^1^	11	20
CS9	*SETBP1*	18	c.3562G>A	p.(Glu1188Lys)	missense	NA ^1^	uncertain	uncertain	0.000079	41	65
CS11	*EXT2*	11	c.1186G>A	p.(Val396Met)	missense	pathogenic	NA ^1^	lik. Benign ^4^	0.00044	17	26
CS11	*DNM2*	19	c.688+2T>C	NA ^1^	spl. don. ^2^	NA ^1^	NA ^1^	pathogenic	NA ^1^	13	16

^1^ NA = not available; ^2^ spl. don. = splice donor; ^3^ lik. path. = likely pathogenic; ^4^ lik. benign = likely benign.

## Data Availability

Publicly available datasets were analyzed in this study. The data can be found at https://www.ncbi.nlm.nih.gov/sra/PRJNA896891.

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
