# Peer review of "Genomic Characterization of Rare Primary Cardiac Sarcoma Entities"

_diagnostics, 2023, doi:10.3390/diagnostics13020214_

Round 1

Reviewer 1 Report

In this research, authors characterised the genomic profile of rare primary cardiac sarcoma entities that are associated with poor prognosis, justified by their scarce information regarding the gene signatures and treatments. The introduction provides a fair background to the subject, but a clear aim must be stated. The experiment design must be better described and it is difficult to follow since there is no clear aim. Bioinformatics analysis needs more explanation in more details. Results and discussion are focused only in the mutation profile. Therefore, I consider the publication of the manuscript after major revisions. Some of the specific recommendations/changes that I suggest are described below:

What is the meaning of “FU” in table 1?

A clear aim is needed in the Introduction section.

In the subsection 2.3 Bioinformatics analysis, why do you cite the web page instead of the publication? Did you use only web-based tools or the command line ones?

The sample size for some of the entities is too small, did you normalise the data to account for the difference in sample size before the PCA, so the comparison can be more robust? You cannot conclude almost anything about the distribution of the entities with this sample size.

What was DNA sequenced for? How did you analyse the DNA data? This is not described in the methods section.

Since you want to characterise the genomic profile of the entities, it would be great to add information on the abundance of the genomic regions and types, percentage of synonymous and nonsynonymous substitutions, number of SNPs, INDELS, etc.

Reviewer 2 Report

In this manuscript, the authors have characterized some features of rare primary cardiac sarcomas. They performed transcriptome and sanger sequencing to analyze their two FFPE samples (one is cardiac osteosarcoma and the other is cardiac leiomyosarcoma). Their studies revealed some knowledge of this rare disease. However, there are some remaining questions to be answered:

1, The authors used some abbreviations. Could the authors spell them out? For example, in Table 1. “Last FU”?

2. The authors discussed the mutation signatures in the manuscript. Since they also performed RNA-seq and performed some general analysis (for example Figure 3), could the authors elaborate more information about their RNA seq results in this manuscript?

Reviewer 3 Report

The author studied the osteosarcoma and leiomyosarcoma cases by applying transcriptome sequencing analyses to enhance the understanding of these cardiac entities and to discover useful information for their clinical management and treatment. Albeit, I consider these findings would guide future clinical interventions to a certain extent, I still have some suggestions.

1, All figures are highly professional, and the authors should guide the readers to the meaning of the images appropriately; otherwise, it is likely to cause misunderstandings. Therefore, I suggest that the author consider revising these figure legends again.

2, The author used whole-transcriptome sequencing on formalin-fixed paraffin-embedded (FFPE) samples from one cardiac osteosarcoma and one cardiac leiomyosarcoma has been performed, to investigate their mutational profiles and to highlight differences and/or similarities with other cardiac histotypes. However, It would be much better if the authors could provide some Workflow or Scheme for this research, I suggest that they can take a look at the recent paper in MDPI (PMID: 35629902, 34834441, 30669548)

3, There are few typo issues for the authors to pay attention to; please also unify the writing of scientific terms. “Italic, capital”? Meanwhile, in the discussion, please also list some limitations that need to be improved in the future.

Round 2

Reviewer 1 Report

All comments were properly addressed and the manuscript is much clearer and well presented.  The reader can now understand the aim and the methods used to achieve the goal. Results and discussion are now broader and more cohesive. Therefore I recommend the publication of the manuscript in the present form.